# Seismic Measurements Using Distributed Acoustic Sensing (DAS) for Underwater Soft Sediment Characterization: Insights from Laboratory- and Field-Scale Measurements

**DOI:** 10.3390/s25237234

**Published:** 2025-11-27

**Authors:** Edwin Obando Hernandez, Matteo Rossi, Roeland Nieboer, Manos Pefkos, Wiebe de Boer, Pieter Doornenbal

**Affiliations:** 1Deltares, 2600 MH Delft, The Netherlands; roeland.nieboer@deltares.nl (R.N.); manos.pefkos@deltares.nl (M.P.); wiebe.deboer@deltares.nl (W.d.B.); pieter.doornenbal@deltares.nl (P.D.); 2Engineering Geology, Lund University, 221 00 Lund, Sweden; matteo.rossi@tg.lth.se

**Keywords:** distributed acoustic sensing, Scholte waves, soft sediments, hydrophone measurements

## Abstract

Scholte wave surveys were conducted at both the laboratory and field scales to evaluate the reliability of distributed acoustic sensing (DAS) with a fiber-optic cable resting on top of unconsolidated sedimentary deposits to determine the distribution of S-wave velocity underneath. Laboratory measurements performed in a controlled environment at the Deltares Laboratory facility demonstrated that DAS retrieves low- and high-frequency energy associated with Scholte and guided waves. The recorded DAS signals provided consistent Scholte wave signals, which depicted coherent phase velocity energy that was used to accurately depict S-wave velocity layering. We observed the presence of guided waves at higher frequencies, which appeared to be enhanced as the source position was moved away from the fiber-optic cable. A field survey was carried out using a linear set-up in a shallow lake, where a fiber-optic cable was placed on top of a sediment layer with a thickness of 5–10 m. The results from DAS were validated using standard hydrophone measurements performed simultaneously. The 2D S-wave velocity cross-section retrieved by DAS appeared to be in good agreement with the results obtained from hydrophone measurements, especially when detecting the main velocity transition occurring at a 7–10 m depth from the free surface.

## 1. Introduction

The energy transition relies on clean energy generated by wind farms, representing a sustainable alternative to fossil fuels. For the construction and maintenance of these wind farms, it is vital to characterize the soil properties properly to determine the bearing capacity required for the engineering design [1,2,3,4]. A parameter often utilized to characterize ground conditions is the small-strain S-wave velocity derived from surface wave surveys composed of Scholte waves that are generated at the water–soil interface [5,6,7]. A Scholte wave survey is carried out similarly to a standard active MASW survey that uses a sledgehammer impact and a linear array of vertical receivers or geophones [8,9]. In offshore seismics, the survey line is towed by a ship, the source consists of an air gun, and the receivers are hydrophones that lie on top of the shallow seabed [10,11,12]. The disadvantages of a standard hydrophone array include the limited number of receivers, which can be an issue when large areas are investigated, and the complex logistics involved when using a large number of receivers.

Distributed acoustic sensing (DAS) technology has emerged as a game-changing technology that offers an unprecedentedly high spatiotemporal resolution for a number of applications, including marine seismic surveying [13,14]. For shallow offshore and onshore subsurface characterization, the advantage of using DAS technology with telecommunication fibers is that it offers the possibility not only of covering very large areas but also of enabling a high resolution for both active and passive surveys [15,16]. However, for offshore surveys, DAS sensors rely on a cable lying on the seabed, which, due to irregularities in the seabed, poses inherent uncertainties, such as limited knowledge of the actual geometry of the fiber and its coupling with the seabed [14,17,18].

The geometry of the fiber-optic cable lying on the seabed is usually mapped through the triangulation of lateral active shots in water, together with previous knowledge of the length of the cable. The coupling conditions are difficult to control because they can vary spatially and over time due to the dynamics of the sediments that may or may not cover the cable [13]. Hence, the lack of sufficient coupling may affect the quality and spatial consistency of the measured seismic waves, posing difficulties in retrieving meaningful information about the soil structure [19]. A comprehensive assessment of the distortions induced due to poor coupling demonstrates that when there is poor contact with the ground, DAS records can be affected by changes in amplitude and phase, which is particularly relevant for surface wave surveys [13,20]. This will have an impact on the coherence of the dispersive waves utilized to determine the near-surface S-wave velocity structure. While there is agreement on the benefits of DAS in mapping the subsurface structure, it is still not well understood as to how the inherent variations in coupling along the fiber affect the accuracy of the S-wave velocity structure underneath the cable.

Results from offshore DAS surveys show clear evidence of the benefits of this technology, particularly due to its larger spatial coverage and faster deployment and measurement time with respect to standard technologies, which are beneficial in offshore environments where weather conditions can change rapidly [16,21,22]. Current efforts have been focused on assessing DAS data quality by testing different cable/fiber types and source configurations in relation to standard measurement systems at localized positions, as well as validating resolved S-wave velocity layering with respect to SCPT (Seismic Cone Penetration Test) and CPT (Cone Penetration Test) point measurements [22]. Less attention has been given to evaluating how effectively DAS recovers the S-wave velocity structure compared to a traditional hydrophone survey, particularly in near-surface characterization. This is particularly relevant due to uncertainties in the accuracy of the 1D S-wave velocity resolved, caused by spatial variations in coupling conditions along the fiber. Furthermore, DAS has been extensively applied to record and analyze Scholte waves in offshore environments, generally focusing on long fiber cables with a low resolution and deep investigations [21,23,24,25].

In this research, we focus on small-scale engineering applications, where the lateral resolution and minimum resolvable depth are key parameters for accurately imaging the distribution of S-wave velocities. We performed laboratory and field measurements using a fiber-optic cable lying on top of a soft sediment layer to investigate how well DAS recovers Scholte waves to resolve the S-wave velocity structure underneath. The laboratory measurements were performed in a controlled environment, meaning that the results could easily be validated by the set-up itself. Field measurements were conducted in a shallow lake utilizing a fiber-optic cable deployed together with a hydrophone array to conduct parallel measurements. The hydrophone measurements were utilized to validate the DAS results. The results are presented in terms of the phase velocity spectrum, utilized to capture the dispersion curve of the fundamental mode of the Scholte waves at several positions along the cable. The measured dispersion curves are later inverted to retrieve S-wave velocity profiles along the survey line. These results demonstrate the reliability of using DAS to retrieve the seismic properties of shallow soft sediment offshore. Ultimately, the goal of this study is to provide a quantitative assessment, based on a combined laboratory- and field-scale approach, of how the similarities of the local S-wave velocity structure recovered with DAS compare to reference ground truth.

## 2. Methods

### 2.1. Wavefield Transformation Method

The wave field transformation method consists of the MASW (Multichannel Analysis of Surface Waves) method developed by Gabriels [26] and Park [8,9,10]. The MASW method is widely used in many engineering applications to obtain information about the subsurface shear wave velocity structure [27]. More recently, high-resolution wavefield transformation techniques such as the frequency-domain beamformer method (FDBFM) [28,29] have been used [30,31,32] in many engineering applications, using both standard geophone and DAS data processing. In this research, the wavefield transformation method implemented was the FDBFM, coded in Python 3.9 [30]. The phase velocity spectrum was used to extract the dispersion curves of the modes of propagation of surface waves. The dispersion curves were input to calculate the S-wave velocity structure via inversion.

Processing and analysis of the measured dispersion curve were performed to determine the minimum and maximum wavelength that could be resolved. The wavelength was defined as the ratio between phase velocity and frequency. The high frequency limit provided information on the shallowest layer thickness to be resolved, while the low frequency limit defined the maximum depth of exploration. As a rule of thumb, we adopted the criteria suggested in the literature [33], which indicates that the minimum and maximum depth of exploration are equivalent to 1/3 of the minimum and maximum measured wavelength, respectively.

### 2.2. Scholte Waves Inversion

The inversion scheme utilized consisted of a least-squares inversion algorithm coded in Python. As both the lake site and the model structure are dominated by a soft-sediment/stiff soil interface, a three-layer finite model was chosen to target the main velocity transition and a potential softer layer at the top of the model. Thus, the inversion was set to determine four S-wave velocities and the thickness of three finite layers. The least-squares scheme utilized a forward model based on Surf96 developed from Computer Programs in Seismology [34]. To account for offshore conditions, the input profile utilized by the forward function was affected by a top layer of density (1000 kg/m^3^) and the Vp (1500 m/s) of water to resemble the water/sediment interface, where Scholte waves propagate [35]. The inversion was focused on the fundamental mode only. Specific considerations, such as the water column and the number of layers utilized for the inversion of laboratory and field data, are explained in later sections.

## 3. Laboratory Experiment

### 3.1. Laboratory Set-Up

To downscale the survey and maintain a high spatial resolution, we utilized a fiber-optic cable wrapped around a PVC pipe. Shot gathers were generated using an active source placed at various positions outside and in line with the DAS survey line, very similar to a standard active surface wave survey. The shot gathers were utilized to retrieve the Scholte waves that occur at the water/sediment interface and that can be used to retrieve the S-wave velocity of the soil structure below the fiber.

The laboratory experiment was performed in the sand flume of the Geohal facility at Deltares’ main campus in Delft. The set-up was built inside a concrete tank 7.0 m in length, 5.5 m in width, and 2.5 m in depth. The sediment properties and geometries were determined through the preliminary modelling of surface waves. We selected a 0.5 m mud/clay layer overlaying a compacted sand layer with a thickness of 1.5 m. Both layers were covered by 0.5 m of fresh water (Figure 1). The measured densities for mud, loose sand, and compacted sand layers were 1372 kg/m^3^, 1568 kg/m^3^, and 1955 kg/m^3^, respectively. The loose sand layer was designed to consider the effect of lateral variation outside the receivers, which is often the case in real-world conditions.

The 2 mm diameter fiber-optic cable wrapped around the PVC pipe (3.3 m in length and 4 inches in diameter) is shown in Figure 2a. The details of the preparation of the PVC for Scholte waves measurements are provided in Table 1. Up to 1650 tightly spaced windings and 578 m of fiber were required to cover the entire PVC pipe. Note that a channel spacing of 5.78 mm was too fine for this application; thus, it was more appropriate to use a coarser sampling to fit the expected minimum wavelength of around 15 cm. Then, the optimum set-up resulted in 57 point sensors (traces) with a channel spacing, dx, of 5.78 cm. This channel spacing (2 × dx = 11.56 cm) was enough to measure a minimum wavelength of 37 cm for both the water column and sediment layers.

This arrangement was designed to provide a very high spatial resolution that matched the dimensions of the set-up. This aspect is relevant because, when using a straight fiber-optic cable, the spatial resolution is constrained by the gauge length of the interrogator. It should be noted that by using a straight cable, the fiber is only sensitive to its longitudinal directional, while with this set-up, the fiber was also sensitive to its transversal direction.

The data recording was performed utilizing an iDAS v2 interrogator, of 10 m gauge length, manufactured by Silixa Ltd. in London, The United Kingdom. To enable the highest possible temporal resolution, the sampling frequency was set to 100 kHz with a duration of 5 s. The spatial resolution for the data acquisition was set to 1.0 m. In post-processing, the number of channels was downsampled to set the optimum channel spacing to 5.78 cm, as described in Table 1. The source was placed at four locations in line with the layout at the top of the sand layer with an increasing distance from the first DAS receiver (s1 = 0.1 m, s2 = 0.5 m, s3 = 1.0 m, and s4 = 1.5 m), as shown in Figure 1. At each source position, five hammer blows were recorded and stacked to improve the SNR (signal-to-noise ratio). The measurements were made utilizing an in-house sledgehammer or “water-hammer” source that consisted of a metallic tube, 0.7 m long, with a rectangular metallic plate attached to one end (Figure 2b). Seismic waves were generated by releasing a rod inside the tube to hit the metallic plate at the bottom. To minimize the potential effect of reflections, a foamy absorbing material was placed on the walls of the tank.

This source was assumed to generate enough broadband frequency energy covering high-frequency guided waves and the low-frequency energy associated with Scholte waves. Although the “water-hammer” was very practical for this lab experiment, further adjustments would be required for implementation in a field-scale survey. Note that the PVC pipe was resting on top of the mud-layer (Figure 2a)—as this was not entirely flat, there may have been some small sections of the fiber (below the PVC pipe) that were not well coupled with the mud layer. Thus, to obtain as uniform contact as possible between the fiber and mud layer, a metallic bar of a similar length was placed inside the PVC pipe, pressing the fiber against the mud.

An example of one of the 5 s DAS records is shown in Figure 3. Each record captured five shots that illuminated the full fiber segment. To minimize the potential effect of vibrations transmitted by the vibration modes of the pipe, the rod in the “water-hammer” was released gently to provide a clear and distinctive shot. Figure 3a shows that together with the main impact, there is also a secondary wave trend generated by the rebounding of the rod, which could not be prevented. Figure 3a shows the wave trends of the main impact between 0.5 and 0.59 s. The shot gathers clearly depict a low velocity trend or move-out that travels along the full pole length.

### 3.2. Dispersion Analysis

An important part of the analysis was determining the measured velocities across a wide frequency range. Figure 4 displays the frequency–wavenumber spectra (f-k) for shot-gathers collected with the hammer source at four locations (s1 to s4) at different distances from the fiber, as presented in Figure 1. The f-k spectra reveal that at all shooting positions, there is an energetic low-velocity event of about 230 m/s which propagates at between 50 and 600 Hz, most likely associated with Scholte waves occurring at the water–sediment interface. This low-frequency velocity is rather prominent at s1 (Figure 4a) and s2 (Figure 4b), but becomes less pronounced at positions s3 (Figure 4c) and s4 (Figure 4d). Then, at position s2 (Figure 4b), a higher-frequency energy trend starts to appear between 2000 and 4000 Hz of about 1500 m/s, associated with the speed of water, yielding wavelengths between 0.375 and 0.75 m. These wavelengths, based on the 1/4 wavelength criterion, yield thicknesses between 0.09 and 0.19 m, which are thinner than the water column of 0.4 m, indicating that trapped or standing waves are not fully reflected back to the free surface, which may be caused by the damping induced by the soft mud layer (with the rigid bottom condition not being fulfilled). This high velocity appears rather prominent when the source moves away from the fiber-optic cable, at positions s3 and s4.

It should be noted that the maximum energy recorded for the Scholte waves decreases when moving the source away from the receivers due to stronger attenuation, especially at higher frequencies. The low-frequency velocities associated with Scholte waves enable the possibility of retrieving the S-wave velocity profile underneath the fiber-optic cable and resolving the mud/compacted sand interface.

Figure 5 displays the phase velocity spectra for all shot-gather positions, focusing on the low-frequency region between 50 and 600 Hz and phase velocities between 50 and 1000 m/s. The spectra in Figure 5 are the same shot records utilized to compute the f-k spectra displayed in Figure 4. It appears that at shot position s1 = 0.1 m (Figure 5a), the energy trends depict a rather consistent fundamental mode up to 360 Hz, which later becomes more erratic. At position s2 = 0.5 m (Figure 5b), there is a similar but much more consistent energy trend up to 500 Hz, which can provide a clear dispersion curve to recover the S-wave velocity profile below the fiber. At positions s3 = 1.0 m (Figure 5c) and s4 = 1.5 m (Figure 5d), the dispersion trend starts to disappear at higher frequencies, making it difficult to interpret a dominant dispersion mode. This is an expected behavior due to inherent energy attenuation, amplified not only by the increased source-to-receiver distance but also due to the very soft sediment/mud layer, which maximizes energy damping, especially at higher frequencies.

For the next step, the inversion was carried out utilizing the dispersion curve interpreted from the phase velocity spectrum displayed in Figure 5b, which provided the clearest and most broadband dispersion trend.

### 3.3. Inverted S-Wave Velocity Profile

We used the phase velocity spectra computed at position s2 = 0.5 m (Figure 5b) to extract the dispersion curve of the fundamental mode. Thus, we emphasize the dominant low- frequency energy trend between 50 and 500 Hz and between 200 and 400 m/s. The dispersion curve yields a minimum wavelength of 0.48 m, which leads to a minimum depth of exploration of 0.16 m. It should be possible to resolve the 0.5 m layer thickness overlaying the compacted sand layer. Note that the main objective of the inversion is to determine to what extent the measured dispersion curve from DAS can approximate the main layer transitions. For this reason, we use the phase velocity spectrum with the most consistent dispersion trend. In this experiment, our true model consisted of a density profile measured once the layering of the laboratory model was finalized.

The inverted dispersion curve and the inverted 1D profile are displayed in Figure 6. The measured (green stars) and calculated (solid blue line) dispersion curves are displayed in Figure 6a, together with the phase velocity spectrum. Figure 6b shows the inverted S-wave velocity profile (blue thick line) plotted together with the density profile (black thick line). The root-mean-squared error (RMSE) of the measured dispersion curves with respect to the inverted dispersion curve is 3.89 m/s. The inverted S-wave velocity profile (blue line) shows two S-wave velocity transitions occurring at depths of −0.6 and −0.96 m. The deeper velocity transition (velocity from 290 to 340 m/s) appears to correspond to the mud/compacted sand interface. This transition at −0.96 m is about 4% off compared to the true interface at −1.0 m, as displayed in the density profile.

## 4. Field Experiment

A similar field-scale experiment was then carried out in a shallow lake environment. In order to mimic the laboratory set-up, a helically wound fiber-optic cable was deployed at the bottom of the lake to record Scholte waves generated by an active source. The fiber-optic cable was deployed together with a 48-hydrophone array to perform parallel measurements using both sensors. The purpose of the hydrophone measurements was to provide validation or reference data to assess the similarities and differences of DAS results with respect to standard systems. This section describes the characteristics of the test site and the field set-up adopted for the measurements with both systems. It presents the results from the dispersion curve analysis, and the inversion results from both fiber-optics and hydrophone measurements. Finally, it provides a quantitative analysis of the differences and similarities of the S-wave velocity profiles recovered from DAS compared to hydrophone measurements.

### 4.1. Site Description

The test site is located in Västersjön lake (Sweden) at the southern edge of the Hallandsåsen mountain ridge, about 20 km from Ängelholm (Figure 7). The lake covers around 460 hectares, with a maximum depth of 12 m, and is the largest lake in the municipality of Ängelholm. The geology is characterized by 5–10 m of quaternary sandy sediments overlaying a crystalline metamorphic rock. The measurement set-up was located at the southern coast of the lake along a straight segment of about 150 m from the coastline (Figure 7). The field survey was carried out between 18 and 19 April 2023.

From a previous survey [36], the measured P-wave velocity structure reveals that at the site (250 m from the coastline), there are three very distinctive velocity structures. In the shallow part, the sediment thickness varies from 5 to 10 m, with an average velocity of about 1700–2000 m/s. Below a depth of 10 m, Vp increases up to 3000–4000 m/s, while at greater depths, the velocity reaches a very high level of up to 5000 m/s. The deeper high-velocity material delineates the interface with the metamorphic bedrock.

### 4.2. Survey Set-Up and Recording Parameters

The survey set-up is described in Figure 8. The seismic survey was carried out using a fiber-optic cable and a hydrophone array deployed along a straight line and as close as possible to each other (~0.2–0.3 m apart, Figure 8a). The fiber-optic cable was an HWC (helically wound cable), 120 m long, with 7 single and 4 multi-mode fibers, contained in three jelly-filled loose tubes (4 fibers per tube), with a wrapping angle of 30 degrees. The HWC was weighed down with a lead line, to ensure, as uniformly as possible, coupling with the lakebed sediments. For the measurements, we used these single-mode fibers with a receiver spacing of 1.0 m. The DAS measurements were performed using the same 10 m gauge length iDAS interrogator utilized in the lab experiment. The output sampling frequency was 5 kHz and the recording duration was 3 s (Figure 8b).

The hydrophone layout consisted of 48 receivers with 2.5 m channel separation, providing a total coverage length of 117.5 m. The hydrophone measurements were carried out using two 24-channel GEODE systems manufactured by Geometrics Inc., in San Jose, CA, USA. The measurements were performed with a sampling frequency of 8.0 kHz and a recorded window of 1.5 s. The source consisted of a Teledyne Bolt 10cu (10 cubic inches) air gun manufactured by Teledyne Bolt in Houston, TX, USA. The GEODE recorders were synchronized with the air gun, so the energy source was fully captured. On the other hand, the source was not synchronized with the iDAS interrogator, and hence the system was triggered manually. The longer recording time of the iDAS interrogator allowed it to fully capture the same shot record as with the GEODE system. These simultaneous measurements allowed us to compare the types of wave measured with both systems to better understand the performance of DAS compared to standard hydrophones.

For the survey, the source was triggered at various positions over the survey line by deploying the air gun from a small boat (Figure 8c). The approximate location of the source is indicated by the green dots. The elevation also indicates that the lake bottom is not flat. Notice that the shots were recorded at irregular intervals due to the weather conditions of strong wind and persistent waves.

### 4.3. Scholte Waves Analysis

The data processing emphasizes distinctive features of both DAS and hydrophone data recorded for the same shot. In this section, the comparisons are carried out using records from both ends and the middle position of the survey line. To indicate the location of the shots, each source position is referred to by the hydrophone number at which the seismic waves appear first. The comparison between DAS and hydrophone shot-gathers is conducted using 24 channels spanning approximately 60 m.

To approximate the hydrophone array configuration (24 hydrophones spaced at dx = 2.5 m), a subset of DAS channels is selected. Specifically, every third DAS channel is chosen, corresponding to a spacing of approximately 2.6 m (3 × 1/1.15). The factor of 1.15 accounts for the helical wrapping of the fiber-optic cable, which affects the actual horizontal distance between measurement points.

Figure 9 displays pairs of DAS and hydrophone records when the source was located at hydrophones numbers 5 (Figure 9a,b), 26 (Figure 9c,d), and 48 (Figure 9e,f). For comparison, the shot-gathers are plotted together with the apparent P-wave and Scholte wave velocities. To enhance the low-frequency energy dominated by Scholte waves, the records are low-pass filtered using a cut-off frequency of 30 Hz. For better visibility of the wave features, the shot-gathers are trace-normalized. Also, all shots use the same horizontal scale to emphasize the comparison of the recorded waveforms retrieved with both systems. At source location 5 (Figure 9a,b), the hydrophone data show very clear wave trends with a dominant apparent velocity of about 280 m/s, which is also visible in DAS data, but from 20 to 60 m. Notice that in the DAS record, there is a high-velocity trend between 0 and 20 m, which is not observed in the hydrophone record. At source location 26 (Figure 9c,d), both the DAS and hydrophone records exhibit similar wave trends that are consistent with the apparent velocity of 280 m/s, but these are better defined in the hydrophone signals. For the shot position near to hydrophone 48 (Figure 9e,f), Scholte waves appear less dominant (due to attenuation) with an increased influence of a high-velocity trend (not visible at the first two positions) which appears very early in the record. This effect seems to be induced by the increased water-column thickness and the distance of the source with respect to the position of hydrophone 48, which seems to be underestimated. The apparently large distance between the receivers and the source enhances the energy of the direct or compressional waves (~1500 m/s) travelling through water. Thus, it is reasonable to assume that the source position was localized around 10–12 m away from the position of hydrophone 48.

The DAS and hydrophones records (displayed in Figure 9) are utilized to calculate the phase velocity spectra at the three locations in a frequency range of 5–30Hz and a phase velocity range of 100–1850 m/s (Figure 10). Likewise, for comparisons at each shot position, the spectra are plotted together with the peak values retrieved from the hydrophone data, meaning that the similarities with respect to DAS can be better visualized. Note that to achieve a fair comparison, before computing the phase velocity spectrum, the DAS shot records are spatially resampled so the channel spacing is set to dx = 2.5 m.

In Figure 10a,b, both the DAS and hydrophone records show that between 8 and 15 Hz, there are similar energy levels, but at higher frequencies, DAS basically does not recover information similar to that observed in the hydrophone data. This lack of higher-frequency energy in the DAS data can be explained by the lower spatial resolution imposed by the 10 gauge length of the interrogator utilized. The hydrophone data, on the other hand, recovers a single and prominent energy trend, which continues up to 25 Hz and can be interpreted as the fundamental dispersion mode. When comparing the DAS energy trend and the extracted peak values of the hydrophone data (10–15 Hz), DAS depicts slightly higher velocities at the same frequencies.

For the second shot position (Figure 10c,d), it appears that both DAS and hydrophones recover similar multimodal features, depicting similarities in frequencies and velocities. Again, the hydrophone data displays more energy at higher frequencies with respect to DAS. For the third shot position (Figure 10e,f), DAS captures part of the energy trend recovered by the hydrophones, specifically between 9 and 12 Hz, while the energy above 12 Hz is shifted to higher modes of propagation. All spectra also show how the source position and the thickness of the water column affect the presence of direct waves travelling into the water column. It is observed that when increasing the depth of the water column below the source position, the direct waves in water (1200–1500 m/s) start to appear at the same frequency range (particularly between 20 and 30 Hz) at which the Scholte waves occur.

### 4.4. Inverted S-Wave Velocity Profiles

The measured dispersion curves associated with the fundamental mode recovered from the phase velocity spectrum are inverted to compute 1D S-wave velocity profiles at 10 locations along the fiber-optic cable. The source positions utilized are relative to hydrophone channel numbers, namely 3, 5, 8, 10, 14, 20, 26, 30, 42, and 48, as shown in Figure 8c. The inversion is performed using the least-squares procedure as described in the Section 2. For the inversion of each position, the thickness of the water column added on the top of the model is obtained from the difference in elevations displayed in Figure 8. The inversion is carried out using three finite layers on top of a half-space. Three finite layers provide enough degrees of freedom to target the main transitions at the site, including the dominant sediment/rock interface. The minimum and maximum S-wave velocities are constrained using the measured phase velocities with a maximum wavelength of around 60 m.

Figure 11 shows the S-wave velocity cross-sections retrieved from inverted 1D S-wave velocity profiles derived from both hydrophones and DAS systems. The 1D profiles for DAS and hydrophones are indicated as D1-D10 and H1-H10, respectively. For both D and H, the numbers 1 to 10 correspond to the 10 source positions utilized to extract the dispersion curves. The misfit range of inverted dispersion curves is 2.7–10.3 m/s (hydrophones) and 2.5–8.5 m/s (DAS). The 2D cross-sections are calculated using kriging interpolation of the 1D inverted S-wave profiles. The 2D S-wave velocity distribution is displayed with the minimum resolvable depth indicated by the black dashed line, which is calculated by dividing the minimum phase velocity by the maximum frequency observed in the dispersion curve at each source position.

Figure 11a shows the S-wave velocity cross-section computed from DAS. It reveals a clear low-velocity layer with S-wave velocities between 180 and 360 m/s and a thickness of about 7–10 m. At the end of the profile, there are a few spots with velocities up to 450 m/s. At a greater depth, there is a higher-velocity layer, ranging between 390 and 600 m/s. The thickness of the top layer (minimum exploration depth) is consistent with the minimum exploration depth curve, which varies between 6.0 and 11.52 m. Figure 11b shows the same S-wave velocity profile calculated using the 1D inverted profiles of the hydrophone data. Again, there is a clear two-layer structure. The top layer has a thickness of about 7–10 m, with a velocity between 150 and 320 m/s, and is on top of a deeper higher-velocity layer (360–720 m/s). In this case, the minimum resolvable depth curve varies between 2.8 and 5.49 m, which indicates that hydrophones not only provide reliable information about the main sediment/rock transition but also shallower velocity variations.

The difference between the two S-wave velocity cross-sections is displayed in Figure 12. The error plot is presented using the mean absolute percentage error (MAPE) throughout the entire cross-section. Here the hydrophone S-wave profiles are taken as the actual/real S-wave velocities. It is observed that, in general, the larger MAPE values (as high as 90%) occur at shallower depths than the minimum resolvable depth derived from hydrophones, which is expected given the lower resolution of DAS. At a greater depth, there is a section with relatively high MAPE values of about 40%, which seems to be underestimated by DAS.

#### Correlation Between DAS and Hydrophone S-Wave Velocities

Provided that both DAS and hydrophone surveys are sampling the same space, it is worthwhile to evaluate the degree of correlation between the S-wave velocity values retrieved from profiles placed at common coordinates. Thus, we use all velocities of profiles from D1 to D10 and from H1 to H10. By using a correlation matrix, it is observed that the DAS and hydrophone velocities have a correlation coefficient of 0.66, meaning that both surveys are similar in about 66% of all retrieved velocities (Figure 13a). By plotting both datasets (after normalizing), it is observed that there is a good correlation which can be utilized to derive the main velocity clusters. By using the k-means clustering method, it is possible to group all velocity values into two clusters (Figure 13b). The number of clusters is determined by the elbow method. For the two clusters, the Silhouette score is about 0.55. Then, for each label, all S-wave velocities are grouped and averaged based on the two clusters identified. These two clusters are assigned to the 1D profiles D1–D10, revealing a well-defined layering that helps to distinguish the main velocity structures (Figure 13c).

The unsupervised classification displayed in Figure 13 can be useful in determining to what extent the inverted profiles from DAS and hydrophones share similar information. The two clusters’ indices are common for both types of data, so by averaging the inverted velocities, it is possible to determine how similar the S-wave velocities are.

Figure 14 shows how the average velocities are propagated using the two clusters or material defined for both DAS and hydrophone velocities. Figure 14a shows that the top layer has an average velocity of 286 m/s, overlaying a stiffer layer with a velocity of 498 m/s. Figure 14b shows the same two-layer distribution for hydrophone data. Here, the top layer has an average velocity of 273 m/s, 4.76% slower than that obtained by DAS, while the deeper layer has an average velocity of 561 m/s, which is about 11% faster than that obtained with DAS.

## 5. Discussion

Laboratory measurements revealed that by placing the fiber-optic cable on top of the soft layer, it is possible to retrieve meaningful Scholte wave energy trends. Although the inverted S-wave velocity profile provided a good estimate of the mud/compacted sand interface, the lack of information on the true velocities precludes validation of the estimated velocities. Note, however, that estimating the S-wave velocity profile by using SCPT or any other invasive method can be very challenging due to the unstable conditions of the mud layer. Despite this limitation, the measured dispersion curve already provides very valuable information on the thickness of the mud layer overlying a compacted sand layer. The wound fiber indeed made it possible to increase the spatial resolution of the survey, allowing us to resolve wavelengths much shorter than those required to resolve the target mud layer. The laboratory experiment certainly provided useful insights on how to downscale the DAS set-up to acquire, regardless of the gauge length of the interrogator, high-spatial-resolution data. Likewise, it was possible to secure proper coupling conditions for the PVC pipe with respect to the soft mud layer. Thus, the main objective of the laboratory experiment was to acquire records with the best possible coupling to resolve the main interface of the layer model. It would be interesting to perform a comparative analysis on how various coupling conditions would affect the quality of the signals in relation to various types of sediment. This aspect, however, could not be addressed in this research, but it should be one of the research areas explored to further develop knowledge on how adverse conditions offshore can affect the performance of DAS.

In order to investigate how the field conditions, such as a non-uniform seabed, affect the DAS measurements, a field experiment was carried out adopting two analogous layouts to perform DAS and hydrophone measurements in parallel. The field survey provided very interesting insights regarding the accuracy of the DAS data with respect to standard hydrophones in real-world conditions. A difference in the DAS field measurements with respect to the lab experiment is that the coupling conditions are more or less controlled by adding extra weight to the fiber-optic cable, while the spatial resolution is constrained by the gauge length of the interrogator. The uncertainties in coupling and gauge length constraints may diminish the quality and vertical resolution of the survey. Of course, in real-world surveys, adequate coupling conditions, as seen in the laboratory experiment, will be difficult to achieve because usually in offshore surveys, the seabed is not entirely horizontal, not to mention the environmental conditions, such as wave movements, and external noise, like vibrations generated by passing ships.

It is important to note that the vertical resolution of DAS is about 2 times lower than that provided by hydrophones, meaning that velocity changes at depths shallower than 6.0 m will not be captured by DAS. Despite the differences in spatial resolution, the DAS succeeded in retrieving the overall velocity structure characterized by a low-velocity sediment layer with a thickness of 7–12 m, overlaying a high-velocity rock formation. It was assumed that hydrophones provide trustworthy information that can be used to somehow validate the DAS results. The S-wave velocities resolved with DAS appeared to be in good agreement with those from hydrophone data. DAS resolved the main layers but not the small variations at shallow depths, which deviated quite a lot from the reference values. This limitation is caused by the lack of energy recorded for the fundamental mode at higher frequencies. This discrepancy in the energy spectra could be explained by the lower spatial resolution of DAS that is imposed by the 10 m gauge length.

Despite the uncertainties in coupling and the differences in spatial resolution, the clustering analysis showed that both DAS and hydrophones capture very similar information on the main velocity structure at the test site. The similarities between DAS and hydrophones could be improved by using, for example, a 2 or 3 m gauge length interrogator; however, this would diminish the quality of the signal in terms of SNR (Signal-to-noise ratio). Another aspect to consider regarding the differences in the recorded signals is that the physical coupling of the cables on top of the sediments varies between the two tested methods. There are thus practical implications when performing DAS surveys in the field, such as the procedure of adding weights to the DAS cable, which helps the fiber to maintain good contact with the sediment layer.

## 6. Conclusions

Both the laboratory and field measurement results demonstrate that despite the uncertainty in the coupling conditions, DAS retrieves Scholte waves that can be utilized to recover at least the overall S-wave velocity distribution. Both experiments demonstrated that a cable lying on top of a shallow soft sediment layer can be utilized to resolve the main velocity transitions. The wound fiber utilized in the laboratory experiment recorded Scholte waves of very high frequencies, allowing a very high spatial resolution to be obtained, up to one-third of the thickness of the shallowest target layer. Besides the low velocity Scholte waves, very high frequencies associated with guided waves travelling alongside the water column were clearly observed.

The results of the DAS field measurements carried out in a lake environment appeared in good agreement with reference data recorded with standard hydrophones. Despite the differences in spatial resolution, both DAS and hydrophone systems provided very similar soil layering, characterized by two distinctive geological units. It was also observed that both datasets are highly correlated and share very similar velocity structures, with differences in average velocity between 5% and 11%. The largest discrepancies between DAS and hydrophone surveys occur at very shallow depths, where the differences in S-wave velocity can be substantial.

## Figures and Tables

**Figure 1 sensors-25-07234-f001:**
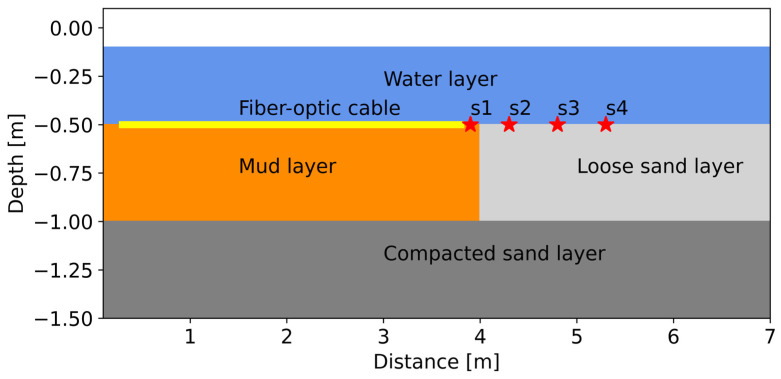
Laboratory set-up for underwater seismic measurements. The DAS channels are represented by a yellow thick line, and the four shot positions are represented by the red stars (s1 = 0.1 m, s2 = 0.5 m, s3 = 1.0 m, and s4 = 1.5 m, from the first DAS receiver at the right-hand side).

**Figure 2 sensors-25-07234-f002:**
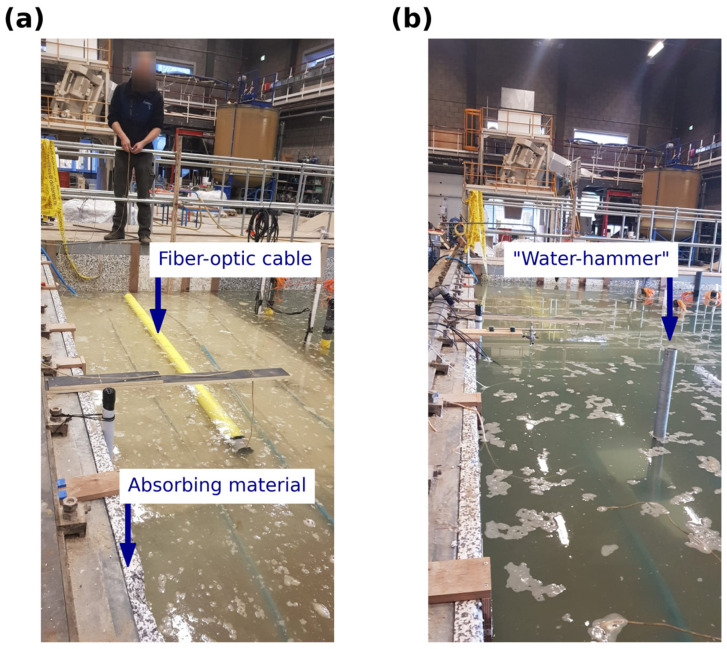
Scholte wave surveys using a “water-hammer” source: (**a**) position of the PVC pipe with a fiber-optic cable wrapped around it (yellow cable) and (**b**) position of the “water-hammer” (vertical hollow tube) in line with the PVC pipe to generate seismic waves.

**Figure 3 sensors-25-07234-f003:**
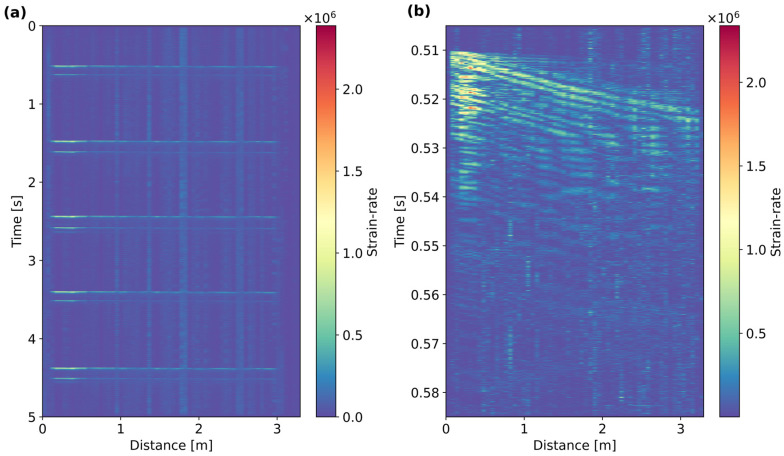
(**a**) Full 5 s DAS record with five shots and (**b**) selected wave trend containing Scholte waves from the first shot in (**a**) between 0.5 and 0.6 s. In (**a**), there are two wave signatures: the main blow and a second smaller-amplitude blow caused by a rebound.

**Figure 4 sensors-25-07234-f004:**
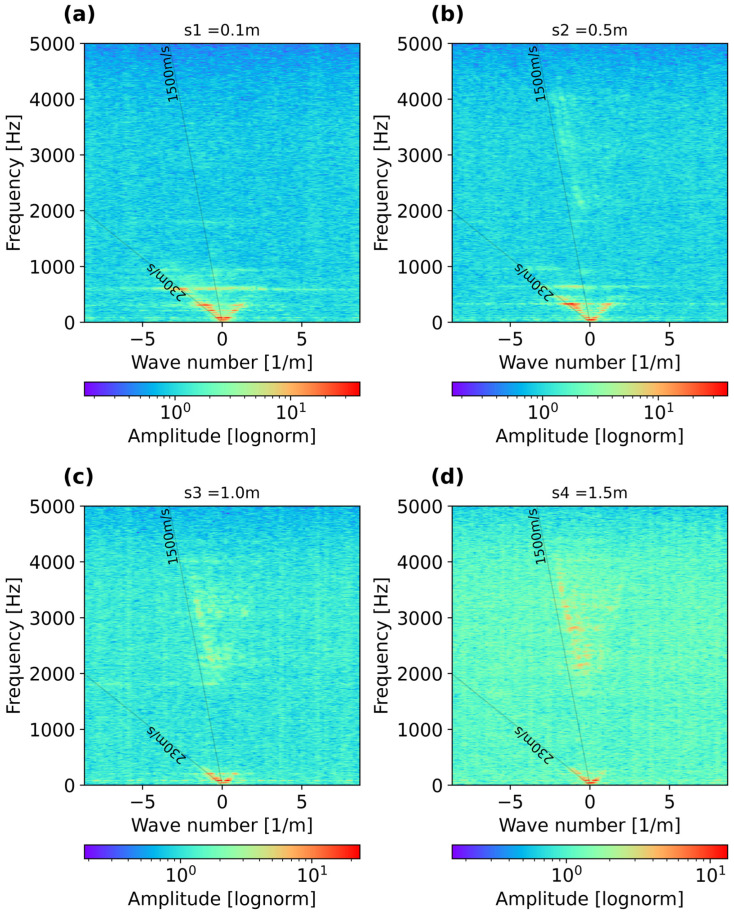
Frequency–wavenumber spectra (f–k) recorded at different shot positions: (**a**) s1 = 0.1 m, (**b**) s2 = 0.5 m, (**c**) s3 = 1.0 m, and (**d**) s4 = 1.5 m.

**Figure 5 sensors-25-07234-f005:**
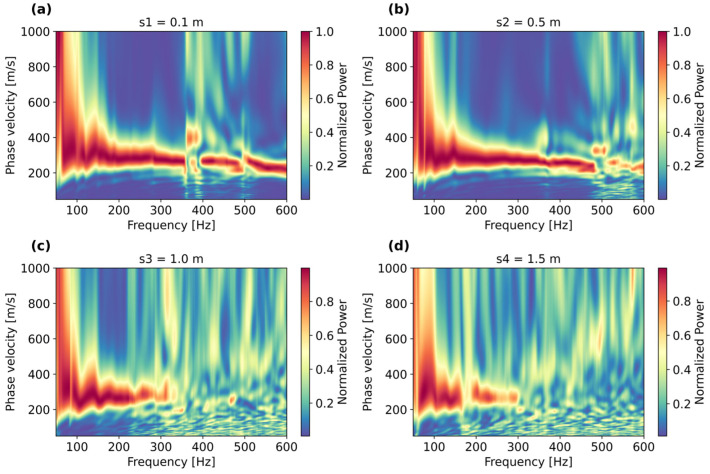
Average phase velocity spectra for source positions at (**a**) s1 = 0.1 m, (**b**) s2 = 0.5 m, (**c**) s3 = 1.0 m, and (**d**) s4 = 1.5 m.

**Figure 6 sensors-25-07234-f006:**
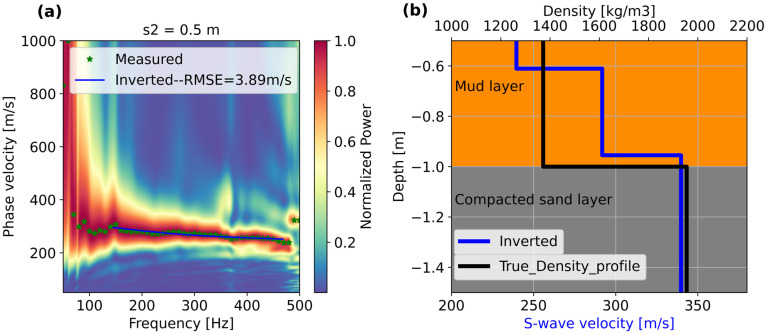
(**a**) Average phase velocity spectrum at position S2 = 0.5 m and (**b**) inverted S-wave velocity profile underneath the fiber. The green stars in (**a**) represent the maximum amplitudes of the phase velocity spectrum.

**Figure 7 sensors-25-07234-f007:**
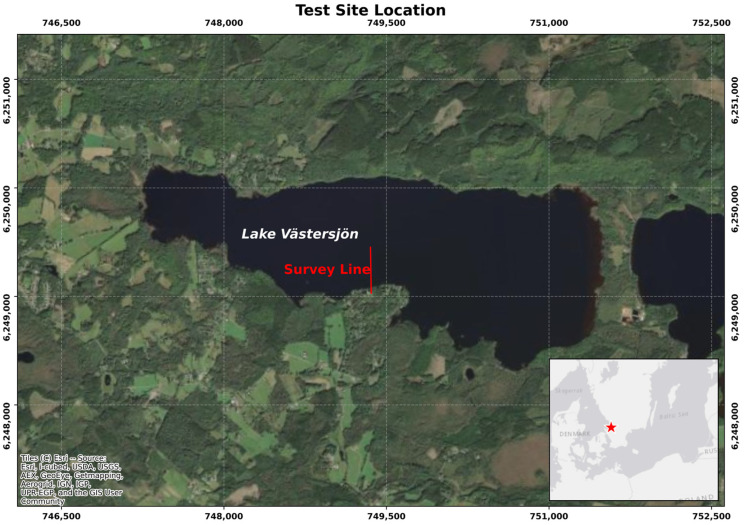
Location map (satellite image from google) of the test site in Sweden. The red line represents the approximate location at which the hydrophone array and DAS cable were deployed. The geographical location of the test site in indicated by the red star.

**Figure 8 sensors-25-07234-f008:**
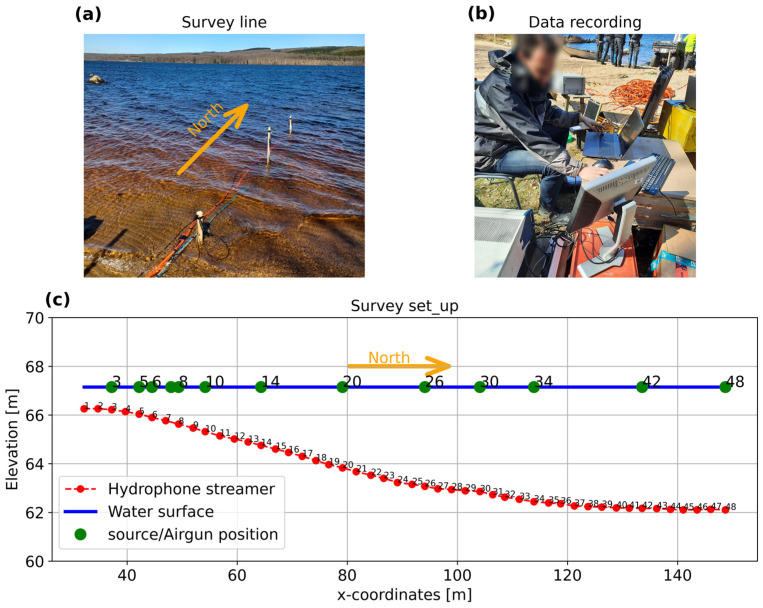
(**a**) Deployed fiber-optic cable (blue cable) alongside the hydrophone streamer (orange cable), (**b**) data acquisition, and (**c**) scheme of the source positions. The blue line represents the water level, the green dots approximate the source positions, and the red dots depict the location of the 48 hydrophones.

**Figure 9 sensors-25-07234-f009:**
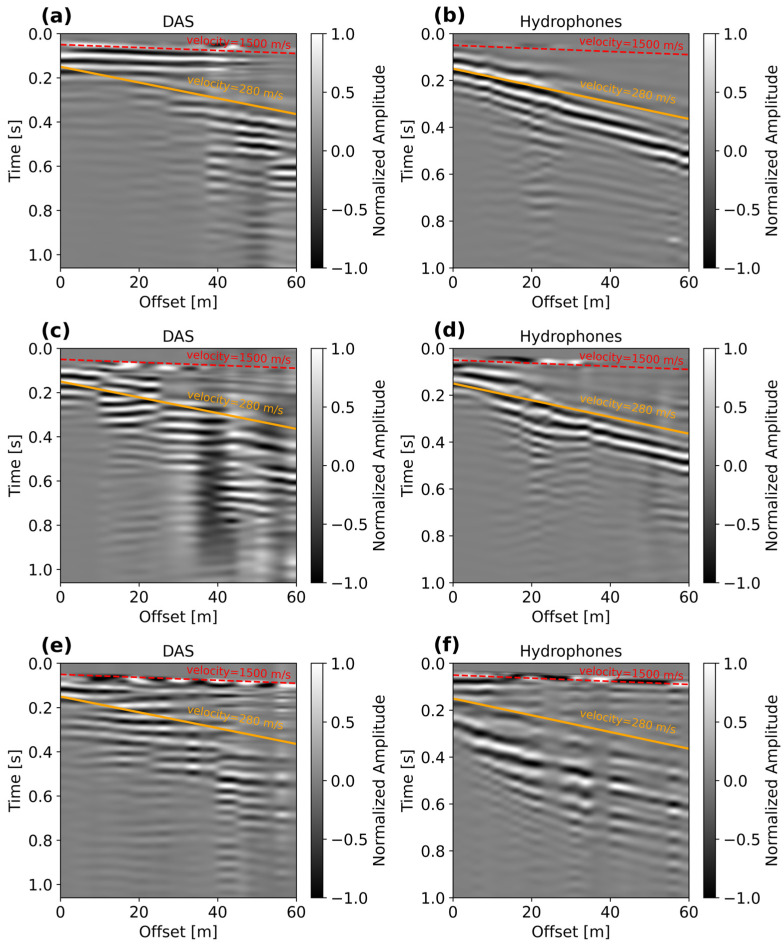
Recorded DAS and hydrophones records are generated using source position with respect to hydrophone channels: (**a**,**b**) 5, (**c**,**d**) 26, and (**e**,**f**) 48. For reference, the speed of water (1500 m/s) and the dominant apparent slow-velocity Scholte waves (280 m/s) are indicated in red and orange, respectively.

**Figure 10 sensors-25-07234-f010:**
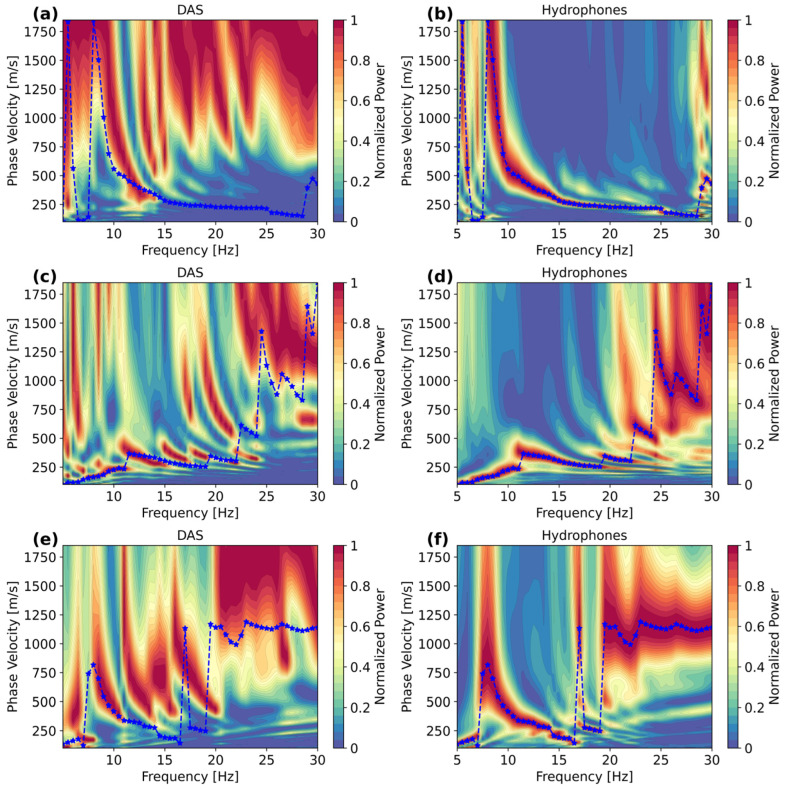
Phase velocity spectra from DAS and hydrophone records with sources at hydrophone numbers: (**a**,**b**) 5, (**c**,**d**) 26, and (**e**,**f**) 48. The blue lines are the selected maxima for each frequency in the hydrophone dataset, which, for comparison, are also plotted on the DAS spectra. The blue stars represent the maximum energy per frequency.

**Figure 11 sensors-25-07234-f011:**
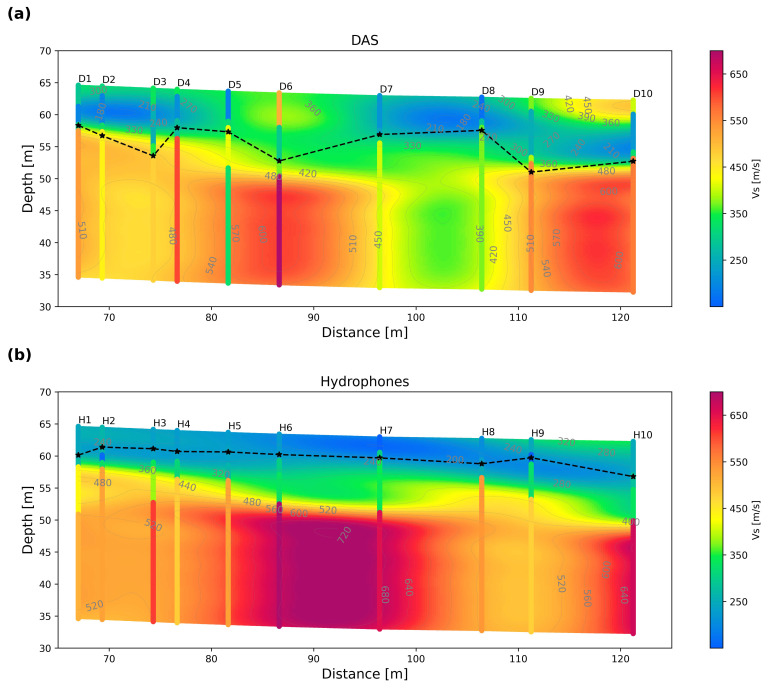
Retrieved S-wave velocity cross-section for (**a**) DAS and (**b**) hydrophone data. The S-wave velocity cross-section is computed after interpolating both DAS and hydrophone 1D profiles, indicated as D1–D10 (DAS) and H1–H10 (hydrophones). The black dashed line represents the minimum resolvable depth based on the minimum wavelength measured. The stars indicate the position of the 1D profiles within the H_min curve.

**Figure 12 sensors-25-07234-f012:**
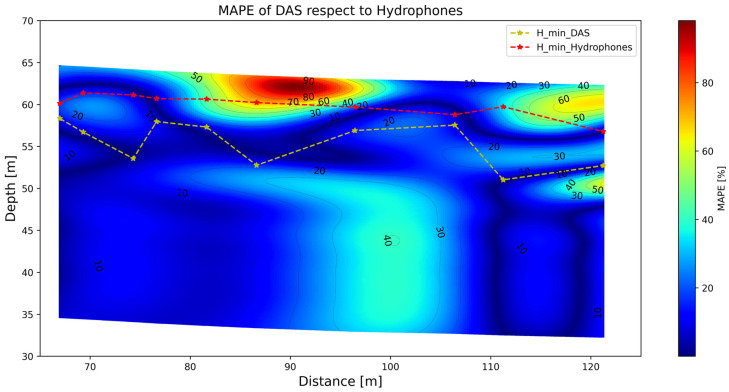
Mean absolute percentage error (MAPE) of the interpolated DAS results compared to the hydrophone inversion. Minimum depth of exploration (H_min) resolved with both DAS and hydrophone systems are indicated by discontinuous red and yellow lines.

**Figure 13 sensors-25-07234-f013:**
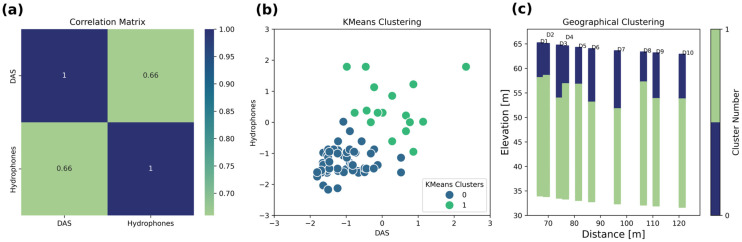
DAS and hydrophone data correlation expressed as (**a**) a correlation matrix, (**b**) k-mans clustering, and (**c**) geographical clustering. The unsupervised classification is performed using the scikit-learn Python library.

**Figure 14 sensors-25-07234-f014:**
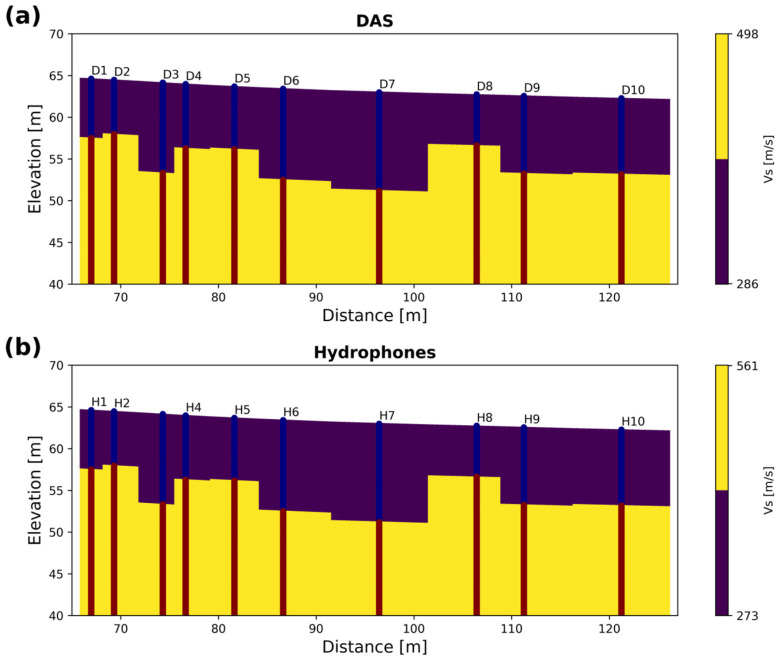
Retrieved S-wave velocity cross-section from averaging inside the geographical cluster for (**a**) DAS and (**b**) hydrophone data. The cross-sections are plotted together with the ten 1D profiles that delineate the main velocity interface. Differences in colors make the 1D profiles distinguishable from the computed cross-section.

**Table 1 sensors-25-07234-t001:** Parameters used to compute the number of DAS channels utilized for lab measurements.

Parameter	Value	Remarks
4″ PVC pipe outer diameter [m]	0.11	Measured value
PVC pipe length [m]	3.3	
Fiber length [m]	578	Minimum required to cover the pipe
Fiber diameter [mm]	2	
Channel spacing [mm]	5.78	This is the horizontal spacing along pipe.
Number of windings	1650	
Number of channels	570	Total number of channels considering 1 m spatial resolution.
Optimum channel spacing [cm]	5.78	After selecting channels every 10 m
Minimum target wavelength [cm]	37	For both sediment and water column layers

## Data Availability

The data used to support the findings of this study are available from the corresponding author upon request.

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
