# Peer review of "Seismic Measurements Using Distributed Acoustic Sensing (DAS) for Underwater Soft Sediment Characterization: Insights from Laboratory- and Field-Scale Measurements"

_sensors, 2025, doi:10.3390/s25237234_

Round 1
Reviewer 1 Report
Comments and Suggestions for Authors
This manuscript presents a well-structured and valuable study comparing Distributed Acoustic Sensing (DAS) with conventional hydrophone arrays for characterizing underwater soft sediments using Scholte waves. The combination of controlled laboratory experiments and field validation in a lake environment provides strong evidence for the reliability of DAS in this application. The manuscript is generally well-written, and the methodology is sound. I recommend acceptance after minor revisions addressing the points below.
- The laboratory setup uses a fiber wrapped around a PVC pipe, which introduces sensitivity in the transverse direction, unlike standard straight DAS cables used in the field. While this is acknowledged as a means to increase spatial resolution for the lab scale, the discussion in Section 5 (Discussion) could more explicitly contrast the implications of this difference for signal quality and coupling compared to the field setup. A sentence or two summarizing how the lab results (especially regarding coupling) might translate, or differ, for a straight cable deployment would be helpful.
-
The manuscript clearly states that the 10 m gauge length limits the high-frequency resolution of the field DAS system. However, it would be beneficial to provide a more quantitative estimate of the resulting vertical resolution limit (e.g., based on the 1/3 wavelength rule) for the field data and explicitly compare it to the resolution achieved by the hydrophones. This would strengthen the discussion on the observed differences in shallow layers (Section 4.4 and 5).
Author Response
Please, see attachment.

Reviewer 2 Report
Comments and Suggestions for Authors
This study employs a Distributed Acoustic Sensing (DAS) system in combination with Scholte wave observations to conduct experimental research, demonstrating strong application potential and methodological innovation. However, the paper still shows several weaknesses in articulating research originality, providing detailed experimental design parameters, and justifying inversion constraints. To enhance the scientific rigor and persuasiveness of the work, I suggest addressing the following points:
- The introduction provides a sufficient overview of the research background but lacks a clear statement of the study’s innovation and core scientific questions. It is recommended that the authors explicitly summarize, at the end of the introduction, how this work improves upon previous studies—for example, in experimental design, inversion accuracy, or DAS signal identification and interpretation methods.
- Although the introduction mentions the limitations of conventional hydrophone arrays and the potential advantages of DAS technology, it lacks a systematic review of recent related studies. The authors are encouraged to supplement the discussion with experimental or numerical studies on the coupling characteristics of DAS and Scholte waves, clarifying how this study differs from and advances existing work, and better defining its academic contribution.
- While the experimental setup is described, key geometric and measurement parameters remain insufficiently detailed. For instance, the fiber winding spacing, number of turns, number and spatial distribution of strain-sensing points on the PVC pipe are not clearly stated. It is suggested that these parameters be provided in a table or schematic to help readers better understand the DAS system configuration and its spatial resolution.
- The manuscript notes that the PVC pipe surface is not completely smooth, which may result in uneven contact between the fiber and the sediment. Such nonuniform coupling could affect the phase continuity and amplitude characteristics of the DAS signal, thereby reducing inversion accuracy. The authors are advised to evaluate this uncertainty through comparative experiments (e.g., adding weights, using filling materials) or numerical simulations to assess its influence on Scholte wave energy distribution and velocity estimation.
- The inversion adopts a “three-layer plus half-space” model, but the rationale behind this layer configuration is not provided. The authors should explain the basis for the layer number selection (e.g., prior geological information, observational resolution limits, or model sensitivity tests).
Author Response
Please, see attachment.

Reviewer 3 Report
Comments and Suggestions for Authors
Review of “Seismic measurements using distributed acoustic sensing (DAS) for under water soft sediment characterization: Insights from laboratory and field scale measurements” by Hernandez et al.
This study validates that the DAS technique can estimate the S-wave velocity structure of sediment layer beneath a fiber-optic cable based on the Scholte wave, using both laboratory and in-situ measurements. Firstly, a dispersion curve is derived from a laboratory experiment. And then, the field experiment is conducted, and the results are compared with those obtained from the co-located hydrophones. Velocity transient zone can be identified at depth of 7 – 10 m below the water-lake floor interface by the DAS. Both the laboratory and field experiments are unique, and their findings are valuable to the marine engineering community. The authors report a high correlation (66%) between DAS and hydrophone observations in the field experiment, suggesting that DAS has strong potential for investigating sediment layer structures. Overall, the presentation is well organized, and the article is suitable for publication once the minor revisions below are addressed.
Line Comments
Lines 111–112: Is the fiber helically wound around the PVC pipe and sufficiently coupled with the bottom material? I’m curious about the difference in sensitivity of the fiber channel between the water layer and the water–bottom material interface.
Lines 174–177: The authors refer to the 1/4 wavelength criterion (or the resonant frequency of the water layer), implying that the expected water depth is too shallow compared to the experimental setup. Isn’t this simply a hydroacoustic wave propagating through the water column?
Figure 7: The copyright information for the background map should be properly cited.
Author Response
Please, see attachment.
